# Longitudinal 8-Epi-Prostaglandin F2-Alpha and Angiogenic Profile Mediator Evaluation during Pregnancy in Women with Suspected or Confirmed Pre-eclampsia

**DOI:** 10.3390/biomedicines12020433

**Published:** 2024-02-14

**Authors:** Anda Lorena Dijmărescu, Florentina Tănase, Marius Bogdan Novac, Mirela Anişoara Siminel, Ionela Rotaru, Daniel Cosmin Caragea, Maria Magdalena Manolea, Constantin-Cristian Văduva, Mihail Virgil Boldeanu, Lidia Boldeanu

**Affiliations:** 1Department of Obstetrics and Gynecology, Faculty of Medicine, University of Medicine and Pharmacy of Craiova, 200349 Craiova, Romania; lorenadijmarescu@yahoo.com (A.L.D.); tina.tanase@yahoo.com (F.T.); magdalena.manolea@umfcv.ro (M.M.M.); cristian.vaduva@umfcv.ro (C.-C.V.); 2Department of Anesthesiology and Intensive Care, Faculty of Medicine, University of Medicine and Pharmacy of Craiova, 200349 Craiova, Romania; 3Department of Neonatology, Faculty of Medicine, University of Medicine and Pharmacy of Craiova, 200349 Craiova, Romania; mirelasiminel@gmail.com; 4Department of Hematology, Faculty of Medicine, University of Medicine and Pharmacy of Craiova, 200349 Craiova, Romania; rodirot@yahoo.com; 5Department of Nephrology, Faculty of Medicine, University of Medicine and Pharmacy of Craiova, 200349 Craiova, Romania; caragea.daniel87@yahoo.com; 6Department of Immunology, Faculty of Medicine, University of Medicine and Pharmacy of Craiova, 200349 Craiova, Romania; 7Medico Science SRL—Stem Cell Bank Unit, 200690 Craiova, Romania; 8Department of Microbiology, Faculty of Medicine, University of Medicine and Pharmacy of Craiova, 200349 Craiova, Romania; lidia.boldeanu@umfcv.ro

**Keywords:** 8-epi-prostaglandin F2-alpha, sFlt-1/PlGF ratio, oxidative stress, angiogenic mediators, pre-eclampsia, hypertension

## Abstract

*Background*: In this exploratory study, we aimed to evaluate the dynamics of angiogenic [soluble fms-like tyrosine kinase-1 (sFlt-1), placental growth factor (PlGF), soluble Endoglin (sEng), and sFlt-1/PlGF, PlGF/sFlt-1, and sEng/PlGF ratios] and oxidative stress [8-epi-prostaglandin F2 alpha (8-epi-PGF2α) and 8-epi-PGF2α/PlGF ratio] mediator levels in women with suspected or confirmed pre-eclampsia (PE) at least two times during pregnancy. We also wanted to identify the possible correlations between 8-epi-PGF2α and angiogenic mediator levels at the time of inclusion of pregnant women. *Methods*: We included 40 pregnant women with suspected or confirmed PE, with a mean age of 29 years (range between 18 and 41 years) and gestational age between 18 and 28 weeks at inclusion in this study. The Enzyme-Linked Immunosorbent Assay (ELISA) method to measure the levels of serum angiogenic and oxidative stress mediators was used. *Results*: The evaluation of baseline sFlt-1/PlGF ratios using a cut-off of 38 suggested that 25 pregnant women had a sFlt-1/PlGF ratio of >38 (sFlt-1/PlGF ratio of >38 group) and 15 had a sFlt-1/PlGF ratio of ≤38 (sFlt-1/PlGF ratio of ≤38 group). The increases in sFlt-1/PlGF ratio in the sFlt-1/PlGF ratio of >38 group were caused by both an increase in sFlt-1 (2.04-fold) and a decrease in PlGF levels (2.55-fold). The 8-epi-PGF2α median levels were higher in the sFlt-1/PlGF ratio of >38 group (1.62-fold). During follow-up after pregnancy, we observed that the mean values of sFlt-1 and sEng and the median values of 8-epi-PGF2α and sFlt-1/PlGF, sEng/PlGF, and 8-epi-PGF2α/PlGF ratios increased directly proportional to gestational age for each measurement time until delivery in both groups. For five women who had a sFlt-1/PlGF ratio ≤38 at inclusion, sFlt-1/PlGF ratio was observed to increase to >38 later in pregnancy. We observed that, in the sFlt-1/PlGF ratio >38 group, baseline 8-epi-PGF2α levels better correlated with angiogenic mediator levels. *Conclusions*: Our study shows that 33.33% of pregnant women evaluated for suspected or confirmed PE with a sFlt-1/PlGF ratio of ≤38 displayed a rise in sFlt-1/PlGF ratio in subsequent weeks. In addition, together with angiogenic mediators, 8-epi-PGF2 α can be utilized as an independent predictor factor to help clinicians identify or predict which pregnant women will develop PE.

## 1. Introduction

Pre-eclampsia (PE), an enigmatic complication of pregnancy, is considered a pregnancy-specific disorder caused by the placenta, present in approximately 5% of all pregnancies, and a main cause of maternal–perinatal morbidity and mortality in the world. After the 20th week of pregnancy, PE is typically identified by the onset of hypertension combined with proteinuria or multi-organ involvement. A subset of individuals with pre-eclampsia may develop hypertension, end-stage renal disease, ischemic heart disease, heart failure, and vascular dementia later in life [1,2,3].

Understanding, predicting, treating, and preventing pre-eclampsia are top concerns in healthcare due to the effects the syndrome has on the mother’s and the baby’s health as well as the long-term aftermath.

Pre-eclampsia biomarkers are molecules found in the serum of expectant mothers that change biologically before the condition manifests clinically. Combinations of two or three biomarkers are being studied since inconsistent and poorly repeatable results are common in the study of individual markers as potential indications for the diagnosis and screening of pre-eclampsia [4,5,6].

Pre-eclampsia is linked to changes in maternal circulation levels of pro- and anti-angiogenic proteins, which have been studied as indicators across gestational age [7]. Placental growth factor (PlGF) and soluble fms-like tyrosine kinase-1 (sFlt-1) are pro-angiogenic and anti-angiogenic factors which circulate in the maternal blood system and are altered in pregnancy complicated by PE [8,9,10,11,12].

The imbalance of sFlt-1 and PlGF leads to the clinical manifestation of PE, and changes in these angiogenic factors are frequently observed before clinical symptoms appear. When PE overlaps with chronic arterial hypertension, the International Society for the Study of Hypertension in Pregnancy (ISSHP) 2021 [2] recommendations consider angiogenic imbalance, as demonstrated by an increase in the maternal sFlt-1/PlGF ratio and a reduction in maternal PlGF levels, to be a diagnostic criterion for de novo PE. According to the guidelines, de novo PE is defined as gestational hypertension associated with uteroplacental dysfunction, which is similar to angiogenic imbalance.

The American College of Obstetricians and Gynecologists (ACOG) [13], several European guidelines [14,15,16], and European Society profiles [17,18] advocate for monitoring the maternal sFlt-1/PlGF ratio to aid in the diagnosis and prediction of the onset of PE.

Another important anti-angiogenic factor is soluble Endoglin (sEng) [19]. Research has indicated increased concentrations of sEng, on average, before the onset of PE, which has led to this factor being considered a promising biomarker for the prediction, diagnosis, and prognosis of PE [20,21,22,23,24,25]. The observation of the kinetics of these biomarkers during pregnancy has been investigated by a number of groups [26,27,28,29,30,31,32,33,34,35].

According to Ishihara et al., higher levels of isoprostanes and prostaglandin metabolites [prostaglandin F2 alpha (PGF2α) metabolites and 8-epi-prostaglandin F2 alpha (8-epi-PGF2α)] near the end of pregnancy indicates the importance of both free radicals and oxidation products catalyzed by cyclooxygenase in the normal biological processes of pregnancy [36]. On the other hand, if we refer to the study of the dynamics of 8-epi-PGF2α during pregnancy, in which measurements should be made at least twice for this marker of oxidative stress, we did not find published studies. In a previously published study [37], in the second trimester of pregnancy and in the postpartum period (three months after delivery), we assessed 8-epi-PGF2α in patients with PE compared to pregnant women without PE. Our preliminary study showed that the expression levels of serum 8-iso-PGF2α in the PE group were higher than that in the PE after-delivery group. Previous studies have evaluated 8-epi-PGF2α and 8-hydroxy-2-deoxyguanosine (8-OHdG) at the time of diagnosis and post-partum [38,39,40,41,42].

In this exploratory study, we aimed to evaluate the dynamics of angiogenic (sFlt-1, PlGF, sEng, and sFlt-1/PlGF, PlGF/sFlt-1, and sEng/PlGF ratios) and oxidative stress (8-epi-PGF2α, 8-epi-PGF2α/PlGF ratio) mediator levels in women with suspected or confirmed PE at least two times during pregnancy. We also wanted to identify the possible correlations between 8-epi-PGF2α and angiogenic mediator levels at the time of inclusion of pregnant women with suspected or confirmed PE.

## 2. Materials and Methods

### 2.1. Study Design and Patient Selection

The University of Medicine and Pharmacy of Craiova’s Committee of Ethics and Academic and Scientific Deontology permitted the authors to conduct this study (no. 134/17 September 2021). This research adhered to all medical ethics guidelines as stipulated in the Helsinki Declaration of 1975, which was revised in 2008.

This secondary analysis was based on a prospective cohort research performed in the Department of Gynecology at the Filantropia Municipal Clinical Hospital in Craiova, Dolj County, Romania, between January and July 2023.

We included in this exploratory study sixty-five pregnant women, aged between 18 and 41 years, with a single pregnancy and suspected or confirmed PE, from 18 weeks of gestation. Each patient signed an informed permission form after being informed about the study beforehand.

The selection criteria for PE patients were patients over the age of 18, gestational age of at least 18 weeks, singleton pregnancies, pregnancies with risk factors for PE, a lack of other pregnancy issues, and informed permission. Exclusion criteria included clinically obvious infections, severe hepatic and renal disease, diabetes, pregnancies with fetal congenital malformations, multifetal pregnancies, hematological dysfunction, immunological diseases, and a lack of informed consent.

At the time of inclusion in the study (visit 1), which varied between 18 and 28 weeks of gestation, the participants were evaluated clinically, socio-demographically, and obstetrically. Then, the pregnant women were monitored at 25, 32, and 39 weeks of gestation until the time of delivery.

Forty of the sixty-five pregnant women completed the study and were included in the final analysis, while twenty-five were lost to follow-up: diabetes mellitus (*n* = 5), chronic kidney disease (*n* = 6), clinically obvious infections (*n* = 4), unwillingness to continue (*n* = 3), and relocation (*n* = 7) (Figure 1).

### 2.2. Diagnosis of PE

The suspicion of PE was based on the following clinical and paraclinical signs: the presence of hypertension, the detection of proteinuria, as well as other symptoms compatible with PE; for example, pain in the right upper quadrant or headache with visual impairment.

PE was defined as de novo hypertension in a previously normotensive woman, often characterized as systolic blood pressure (SBP) >140 mm Hg and/or diastolic blood pressure (DBP) >90 mm Hg after 20 weeks of gestation, and significant proteinuria [protein-to-creatinine ratio (PCR) ≥30 mg/mmol, or protein excretion of at least 0.3 g/L in a complete 24 h urine collection or ≥2+ by urinary dipstick if confirmatory testing is not available) that resolves fully by the sixth postpartum week, according to ACOG [1] and ISSHP criteria [2]. Hemolysis, increased liver enzymes, and a low platelet count were all considered symptoms of HELLP syndrome.

#### 2.2.1. Diagnosis of Hypertension

The initial blood pressure measurements were made in a clinical context (clinic or hospital). Their feet were placed on the floor, the cuff was adjusted to their 33 cm middle arm circumference, and the initial measurements were made in both arms. Their blood pressures were taken following a routine procedure. A minimum of two measurements were averaged to compute the blood pressure, which was then verified after an additional 15 min of monitoring.

Pregnant women diagnosed with hypertension were advised to monitor their blood pressures at home once a week and to use a clinically validated device for use in pregnancy and pre-eclampsia (a device accessible to all pregnant women was recommended; for example, Microlife WatchBP Home S, Microlife AG, Widnau, Switzerland). It was also requested that pregnant women measure their blood pressure at least twice a day until delivery.

#### 2.2.2. Determination of Proteinuria

Pregnant women were instructed to collect 10–15 mL of urine in sealed sterile containers first thing in the morning. Proteinuria was determined using the semi-quantitative color scale method using a urine reagent dipstick. We can categorize proteinuria as absent, trace, or having a quantity of 0.3 g/L, 1.0 g/L, or 3.0 g/L, respectively, using the semi-quantitative approach with the strips; this is equivalent to a negative result, the existence of traces, or a positive result scored as 1+, 2+, or 3+. A positive test was defined as ≥0.3 g/L (+1).

#### 2.2.3. Obtaining Blood Samples for Angiogenic Profile and Oxidative Stress Mediator Assessment

The first samples were collected during visit 1 when pregnant women were enrolled in the study.

Because our study aimed to evaluate the evolution of angiogenic (sFlt-1, PlGF, sEng, and sFlt-1/PlGF, PlGF/sFlt-1, and sEng/PlGF ratios) and oxidative stress (8-epi-PGF2α, 8-epi-PGF2α/PlGF ratio) mediator levels in women with suspected or confirmed PE, determinations were made at least two times during pregnancy.

Thus, for example, if a pregnant woman was included in the study at 18 weeks (18w) of pregnancy, the aforementioned determinations were made during visit 2 (25w, 25 weeks of pregnancy), visit 3 (32w, 32 weeks of pregnancy), and at visit 4, if applicable (39w, 39 weeks of pregnancy).

Using simple vacutainers (Vacutest Kima, Arzegrande, Padova, Italy), a 5 mL venous blood sample was taken in the morning from each pregnant patient. A standard methodology was followed to separate the clot by centrifugation (Hermle AG, Gosheim, Baden-Württemberg, Germany) at 3000× *g* for 10 min, no later than 4 h after harvesting.

Each patient’s serum sample cryotubes were coded, sealed to avoid contamination, and maintained at a temperature lower than −80 °C to enable extended sample processing. Until working with the patient samples, the frozen samples were kept at room temperature to avoid freezing–unfreezing cycle activities.

### 2.3. Angiogenic and Oxidative Stress Mediator Assessment

We employed the Enzyme-Linked Immunosorbent Assay (ELISA) method at the University of Medicine and Pharmacy of Craiova’s Immunology Laboratory to measure the levels of serum angiogenic and oxidative stress mediators.

Commercial tests for every mediator were supplied by the manufacturer, Elabscience (Houston, TX, USA): 8-epi-PGF2α (Cat. No.: E-EL-0041; sensitivity: 9.38 pg/mL; detection range: 15.63–1000 pg/mL), sFlt-1 (Cat. No.: E-EL-H6117; sensitivity: 4.69 pg/mL; detection range: 7.81–500 pg/mL), PlGF (Cat. No.: E-EL-H1555; sensitivity: 9.38 pg/mL; detection range: 15.63–1000 pg/mL), sEng (Cat. No.: E-EL-H6010; sensitivity: 0.38 ng/mL; detection range: 0.63–40 ng/mL).

The manufacturer’s instructions and recommended procedures were followed when performing the dilutions and working processes. The procedures made use of a standard optical analyzer (Asys Expert Plus UV G020 150 Microplate Reader, ASYS Hitech GmbH, Eugendorf, Austria) with a 450 nm wavelength.

### 2.4. Statistical Analysis

GraphPad Prism Version 5 (LLC, San Diego, CA, USA) was used to analyze the data. Using the D’Agostino and Pearson omnibus normality test, the data were examined for normality. The distribution of sFlt-1, PlGF, and sEng was normal, and their mean values are shown together with the standard deviation (SD). The non-normal distribution of 8-epi-PGF2α, sFlt-1/PlGF, PlGF/sFlt-1, and sEng/PlGF levels and 8-epi-PGF2α/PlGF ratios was observed, and the results are displayed as the median with inter-quartile range. Percentages were used to express the category values.

To assess the difference between groups, continuous variables were examined using the Mann–Whitney U test or the Kruskal–Wallis H test (used for non-Gaussian distributions). The existence of significant correlations between the levels of sFlt-1, PlGF, sFlt-1/PlGF, sEng, and 8-epi-PGF2α was evaluated using Spearman’s coefficients (−1 < rho < 1).

## 3. Results

### 3.1. Baseline Clinical Features, Profile Mediators, and Diagnosis at the Time of Inclusion

We included 40 pregnant women with suspected or confirmed PE, having an average age of 29 years (range between 18 and 41 years) and gestational age between 18 and 28 weeks at inclusion in the study. Also, at the time of inclusion, twenty-five (62.50%) women were suspected of having PE, ten (25%) women had gestational hypertension, and another ten (25%) women had PE/HELLP syndrome.

After determining sFlt-1 and PlGF levels and the sFlt-1/PlGF ratio from the samples collected during visit 1, depending on the sFlt-1/PlGF ratio cut-off value of 38 suggested by Zeisler et al. [30], 25 pregnant women had a sFlt-1/PlGF ratio cut-off value of >38. In Table 1, we highlight the clinical and paraclinical parameters, baseline angiogenic profile, oxidative stress mediators, and diagnosis at the time of inclusion for all pregnant women, for pregnant women with a sFlt-1/PlGF ratio cut-off value of ≤38, and pregnant women with a sFlt-1/PlGF ratio cut-off value of >38.

Women with a sFlt-1/PlGF ratio of >38 had significantly higher mean levels of SBP (*p* = 0.001), DBP (*p* = 0.036), ALT (*p* = 0.010), and creatinine (*p* = 0.008) compared to women with a sFlt-1/PlGF ratio of ≤38.

#### Baseline Angiogenic Profile and Oxidative Stress Mediators

A Mann–Whitney U test was run to determine if there were differences in baseline angiogenic profile and oxidative stress mediators between the two groups of pregnant women at the time of inclusion (Table 1).

The increases in the sFlt-1/PlGF ratio in the sFlt-1/PlGF ratio of >38 group were caused by both an increase in sFlt-1 (2.04-fold) and a decrease in PlGF levels (2.55-fold); this difference between groups was statistically significant, where U = 106.0, *p* = 0.024, and U = 37.0, *p* < 0.0001, respectively.

Comparing the mean values of sEng, we found that there were no statistically significant differences between the groups.

Median 8-epi-PGF2α levels were higher in the sFlt-1/PlGF ratio of >38 group (1.62-fold). Also, the median values of the PlGF/sFlt-1, sEng/PlGF, and 8-epi-PGF2α/PlGF ratios showed a statistically significant increase in the sFlt-1/PlGF ratio of >38 group compared to the sFlt-1/PlGF ratio of ≤38 group.

### 3.2. Angiogenic and Oxidative Stress Mediator Levels during Pregnancy Follow-Up

In Table 2, we highlight the dynamics of the mean, respective to the median values, during pregnancy in the two studied groups.

During pregnancy follow-up, we observed that the mean values of sFlt-1 and sEng levels, and median values of 8-epi-PGF2α levels and sFlt-1/PlGF, sEng/PlGF, and 8-epi-PGF2α/PlGF ratios, increased directly proportional to gestational age for each measurement time until delivery in both groups.

Also, Figure 2 shows the sFlt-1 and PlGF levels, sFlt-1/PlGF ratio, sEng and 8-epi-PGF2α levels, and other ratios (PlGF/sFlt-1, sEng/PlGF, and 8-epi-PGF2α/PlGF) of individual patients during pregnancy follow-up. We found that in 66.67% (10 patients) of pregnant patients with a sFlt-1/PlGF ratio ≤38 at inclusion [median 21.22 (range, 7.39–36.77)], the sFlt-1/PlGF ratio remained stable for at least 49 days, with a median change per day of 0.62.

#### 3.2.1. Dynamics of Mediator Levels during Pregnancy Follow-Up in the sFlt-1/PlGF Ratio ≤38 Group

For five women who had a sFlt-1/PlGF ratio ≤38 at inclusion (Figure 2C), the sFlt-1/PlGF ratio was observed to increase to >38 during pregnancy follow-up (from 22.52, 25.82, 4.24, 27.45, and 11.88, to 104.22, 79.04, 40.38, 203.66, and 81.77, respectively). The intervals between measurements for these women were 49, 35, 49, 42, and 70 days, and they delivered at 38.5, 39.1, 39.0, 38.4, and 38.3 weeks of gestation, respectively.

Increases in the sFlt-1/PlGF ratio were induced by increases in sFlt-1 (2.50-fold, 1.65-fold, 3.81-fold, 2.05-fold, and 1.35-fold, respectively) as well as decreases in PlGF (1.50-fold, 1.40-fold, 2.50-fold, 1.98-fold, and 5.10-fold, respectively) levels.

Levels of the anti-angiogenic protein sEng were increased in individual patients during pregnancy follow-up for each measurement time until delivery, as can be seen in Figure 2D.

Also, in the case of 8-epi-PGF2α, when we analyzed the dynamics of its levels in individual patients during pregnancy follow-up, we observed an increase directly proportional to gestational age for each measurement time until delivery (Figure 2E).

Regarding the gestational age at which delivery occurred, in this group, it was between 37 weeks and 5 days and 39 weeks and 3 days.

#### 3.2.2. Dynamics of Mediator Levels during Pregnancy Follow-Up in the sFlt-1/PlGF Ratio >38 Group

In this group, the gestational age at which delivery occurred was between 37 weeks and 2 days and 39 weeks and 5 days.

In all cases in the sFlt-1/PlGF ratio >38 group, which at inclusion had a median sFlt-1/PlGF ratio of 76.00 with a range of 39.38-227.02, the ratio remained above the cut-off value of 38, increasing by 2.37-fold at 25w, 5.51-fold at 32w, and 5.58-fold at 39w. For 9, cases we observed a 1.05-, 1.17-, 1.94-, 1.48-, 1.47-, 1.64-, 1.37-, 1.36-, and 1.43-fold decrease in the ratio, respectively; however, it always remained above 38. Decreases in the sFlt-1/PlGF ratio were induced by decreases in sFlt-1 as well as decreases in PlGF levels (Table 2).

The median values of 8-epi-PGF2α levels increased considerably, doubling at each measurement and showing a 9.65-fold increase at 39w.

### 3.3. 8-epi-PGF2α Serum Levels Associated Positively with Angiogenic Mediator Levels at Baseline

At baseline, in the sFlt-1/PlGF ratio ≤38 group, the results showed a statistically significant correlation between PlGF and the sFlt-1 levels (strong negative correlation, rho = −0.754, *p* = 0.001) (Table 3).

sFlt-1 levels also were positively correlated to the limit of significance with 8-epi-PGF2α (weakly correlation, rho = 0.388, *p* = 0.053) and the sEng levels (moderate correlation, rho = 0.466, *p* = 0.056).

8-epi-PGF2α values were weakly positively correlated with sFlt-1/PlGF ratio (rho = 0.219, *p* = 0.046).

In the sFlt-1/PlGF ratio >38 group, the baseline findings showed that 8-epi-PGF2α levels better correlated with angiogenic mediator levels.

8-epi-PGF2α levels correlated strongly and significantly with sFlt-1 (rho = 0.724, *p* < 0.0001) and sEng levels (rho = 0.675, *p* < 0.0001), and moderately and significantly with sFlt-1/PlGF ratio (rho = 0.529, *p* = 0.037).

Regarding sEng levels, a strong statistically significant positive correlation with sFlt-1 levels (rho = 0.669, *p* < 0.0001) was found (Table 4).

## 4. Discussion

In this exploratory study, we aimed to evaluate the dynamics of angiogenic (sFlt-1, PlGF, sEng, and sFlt-1/PlGF, PlGF/sFlt-1, and sEng/PlGF ratios) and oxidative stress (8-epi-PGF2α, 8-epi-PGF2α/PlGF ratio) mediator levels in women with suspected or confirmed PE at least two times during pregnancy.

Using the recommended sFlt-1/PlGF ratio cut-off value of ≤38 [20] to rule out PE, we found that sFlt-1/PlGF ratio did not increase during follow-up in 10 of the 15 patients with suspected or confirmed PE, and remained stable for at least 49 days, with a median change per day of 0.62. For five women with a sFlt-1/PlGF ratio ≤38 at inclusion, sFlt-1/PlGF ratio was observed to increase to >38 during pregnancy follow-up. We found that these pregnancies were complicated by gestational hypertension, intrauterine growth restriction, as well as HELLP syndrome, but they delivered at 38.5, 39.1, 39.0, 38.4, and 38.3 weeks of gestation, respectively.

Studies have observed that monitoring maternal sFlt-1 and PlGF levels after a PE diagnosis may aid in predicting the time between PE diagnosis and delivery, easing the application of clinical care options. Zeisler et al. [43] investigated the relationship between the sFlt-1/PlGF ratio and the time to delivery in an observational cohort study, stating that pregnant women with a sFlt-1/PlGF ratio of >38 had a 2.9-fold higher likelihood of an impending birth compared to those with a lower sFlt-1/PlGF ratio if they had suspected PE between 24 + 0 and 36 + 6 weeks of gestation.

The time interval between diagnosis and delivery in individuals with PE has been examined in a number of studies by conducting consecutive assessments of the sFlt-1/PlGF ratio following a diagnosis. Baltajian et al. [27] carried out observational research in which the sFlt-1 and PlGF concentrations of pregnant patients with suspected PE who were admitted to the hospital were checked once a week until their babies were delivered. They discovered that the rate of increase in anti-angiogenic state was greater in women who had an adverse outcome compared to those who did not. From admission to delivery, Schaarschmidt et al. [26] measured sFlt-1 and PlGF levels in women with confirmed early-onset PE or late-onset PE. Women with early-onset PE experienced a statistically bigger daily increase in sFlt-1, a non-significantly greater daily reduction in PlGF, and a statistically greater daily increase in sFlt-1/PlGF ratio compared to those with late-onset PE. Peguero et al. [44] assessed sFlt-1 and PlGF levels in women with confirmed early-onset PE at admission and immediately before birth. They found that longitudinal increases in sFlt-1 and sFlt-1/PlGF ratio values were more apparent in pregnancies with early-onset severe PE vs. uncomplicated pregnancies, and the median duration from admission to delivery was considerably shorter. Meler et al. [45] discovered extremely low PlGF levels (12 pg/mL) in 87.5% of women diagnosed before 28 weeks, 78.4% of women diagnosed between 28 and 32 weeks, and 41% of women diagnosed after 32 weeks.

In a study conducted by Saleh et al. [46], the authors aimed to assess the evolution of the sFlt-1/PlGF ratio in women with suspected or confirmed PE and to investigate the changes in sFlt-1 and PlGF levels in women diagnosed with PE or HELLP syndrome. It was shown in this study that the sFlt-1/PlGF ratio can continue to rise over several weeks or months, which calls for regular evaluation. Furthermore, in most patients with a high (>38) sFlt-1/PlGF ratio at admission, the sFlt-1/PlGF ratio doubles every week. This study also revealed that, among women with PE and a high sFlt-1 level and sFlt-1/PlGF ratio, these parameters decline rapidly postpartum to values seen in age-matched, healthy non-pregnant women.

Similar to Saleh’s study, in our study, during pregnancy follow-up, we found that the mean values of sFlt-1 levels and sFlt-1/PlGF ratio increased directly proportional to gestational age for each measurement time until delivery in both groups. Increases in the sFlt-1/PlGF ratio were induced by increases in sFlt-1 as well as decreases in PlGF levels. In both groups, sFlt-1/PlGF ratio ≤38 and sFlt-1/PlGF ratio >38, sFlt-1 levels increased further in most cases, almost doubling roughly every measurement, with a mean change per day of 17.59, and 22.67, respectively.

In our opinion, this is the first study to evaluate the dynamics of 8-epi-PGF2α at least two times during pregnancy. In our study, we were even able to obtain three measurements during pregnancy follow-up in twenty-three of the forty cases. As in the case of sFlt-1, the median values of 8-epi-PGF2α in both groups increased further in most cases, almost doubling roughly every measurement, with a mean change per day of 1.37 (sFlt-1/PlGF ratio ≤38 group) and 2.04 (sFlt-1/PlGF ratio >38), respectively.

Studies published by Turpin et al., Sakyi et al., and Anto et al. [38,39,40,42] evaluated the association of oxidative stress biomarkers (8-epi-PGF2α, 8-OHdG, and total antioxidant capacity) and angiogenic growth mediators (sFlt-1, PlGF, sEng, vascular endothelial growth factor-A, VEGF-A) in early-onset pre-eclampsia (EO-PE) and late-onset pre-eclampsia (LO-PE) at the time of diagnosis and 48 h post-partum. Turpin et al. [38] and Alahakoon et al. [11] discovered that EO-PE women had significantly higher angiogenic imbalance and OS than LO-PE women, implying that EO-PE pregnancies are frequently more severe and have worse maternal outcomes than LO-PE pregnancies.

For the first time, Sakyi et al. [39] determined that, in the Ghanaian population, the ratios of 8-epiPGF2α/PlGF and PIGF/8-epiPGF2α were significant diagnostic markers for EO-PE (<34 weeks), with a threshold of 7.2 and above for the former and 0.14 and lower for the latter. PE women were 1.74 times more likely to have EO-PE at this threshold value. Additionally, they reported that sFlt-1/PIGF ratio was not as effective in predicting EO-PE at the threshold value as 8-epiPGF2α/PIGF ratio, but that the former provided a superior diagnostic value for EO-PE (2.69 times vs. 1.74 times).

The increase in oxidative stress interacts with angiogenic growth mediators and influences changes in placental angiogenesis in PE, which unbalances the ratio between angiogenic growth mediators and oxidative stress biomarkers among PE women, an imbalance also found in our study. The combined evaluation of angiogenic growth mediators and oxidative stress biomarkers can contribute not only to diagnosis, prognosis, and therapeutic scope but also to a better understanding of the complexity of PE mechanisms. The results detailed above are supported in our study by the strong and moderately significant correlations obtained between 8-epi-PGF2α levels, sFlt-1 and sEng levels, and sFlt-1/PlGF ratio.

Following this exploratory study, we were able to observe that the dynamic determination of these biomarkers—at least two determinations during pregnancy—provide us with important information about changes in the levels of 8-epi-PGF2α and sEng, along with the well-known biomarkers sFlt-1 and PlGF, during pregnancy.

We recognize that the descriptive character of our study, the limited sample size, and the fact that the patients were recruited from a single clinical university hospital in Craiova provide certain limitations to the analysis.

The observations obtained in our study would certainly benefit from an additional extension of the study, with multicenter involvement. Further study on this area may yield valuable insights into identifying which serum biomarkers might serve as additional therapeutic targets in clinical practice.

## 5. Conclusions

Our exploratory study shows that 33.33% of pregnant women evaluated for suspected or confirmed PE with a sFlt-1/PlGF ratio of ≤38 displayed a rise in sFlt-1/PlGF ratio in subsequent weeks. In addition, together with angiogenic mediators, 8-epi-PGF2α can be utilized as an independent predictor factor to help clinicians identify or predict which pregnant women will develop PE.

## Figures and Tables

**Figure 1 biomedicines-12-00433-f001:**
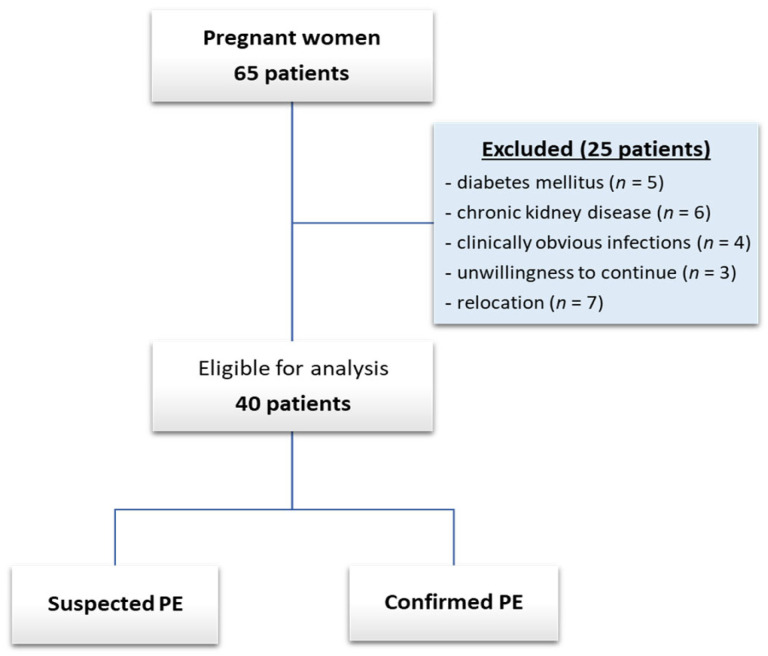
Patient inclusion flow chart.

**Figure 2 biomedicines-12-00433-f002:**
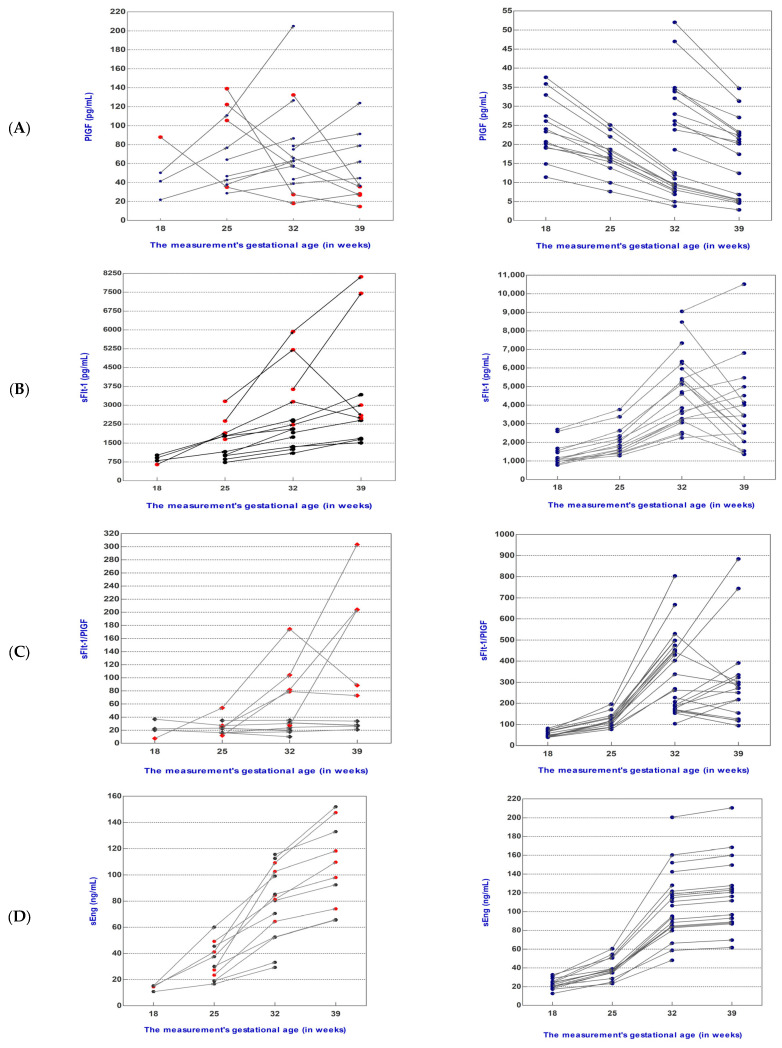
Placental growth factor (PlGF) serum levels (**A**), soluble fms-like tyrosine kinase-1 (sFlt-1) serum levels (**B**), sFlt-1/PlGF ratio (**C**), soluble endoglin (sEng) serum levels (**D**), and 8-epi-prostaglandin F2-alpha (8-epi-PGF2α) serum levels (**E**) were measured in forty women who had a sFlt-1/PlGF ratio of ≤38 (*n* = 15, left) or >38 (*n* = 25, right) at presentation and at different points in their pregnancies. Data for five patients for whom the sFlt-1/PlGF ratio increased from ≤38 at the time of presentation to >38 later in pregnancy are indicated as red squares or rhombuses points (left). Suspected or confirmed pre-eclampsia with a sFlt-1/PlGF ratio of ≤38 (left) or >38 (right) at presentation; (left) sFlt-1/PlGF ratio of ≤38, (*n* = 15); (right) sFlt-1/PlGF ratio of >38, (*n* = 25).

**Table 1 biomedicines-12-00433-t001:** Baseline clinical and paraclinical features, angiogenic profile, oxidative stress mediators, and diagnosis at the time of inclusion of pregnant women with suspected or confirmed PE.

Baseline Parameter	All Pregnant Women (*n* = 40)	sFlt-1/PlGF Ratio of ≤38 (*n* = 15)	sFlt-1/PlGF Ratio of >38 (*n* = 25)	*p*-Value
Clinical and paraclinical features				
Maternal age (years) [median (range)]	29 (18–41)	32 (20–41)	28 (18–36)	0.038 *
Gestational age (years) [median (range)]	24 (18–28)	23 (18–28)	24 (18–28)	0.287 ns
History of PE	6 (24%)	-	6 (24%)	-
Pre-existing hypertension	7 (17.5%)	3 (20%)	4 (16%)	-
SBP [mmHg] (mean ± SD)	143.00 ± 14.90	132.00 ± 15.80	149.00 ± 8.40	0.001 *
DBP [mmHg] (mean ± SD)	89.70 ± 7.80	85.30 ± 6.10	92.40 ± 7.50	0.036 *
Platelets [10^9^/L] (mean ± SD)	248 (147–368)	252 (165–322)	212 (147–368)	0.589 ns
AST [U/L] (mean ± SD)	42.50 ± 20.50	28.10 ± 7.58	51.10 ± 21.10	<0.0001 **
ALT [U/L] (mean ± SD)	46.90 ± 21.10	31.90 ± 11.60	55.90 ± 20.40	0.010 *
Creatinine [mg/dL] (mean ± SD)	0.77 ± 0.13	0.68 ± 0.12	0.83 ± 0.11	0.008 *
Urea [mg/dL] (mean ± SD)	30.80 ± 8.59	26.50 ± 7.41	33.30 ± 8.36	0.054 ns
Baseline angiogenic profile and oxidative stress mediators				
sFlt-1 [pg/mL] (mean ± SD)	2586.00 ± 2182.00	1565.00 ± 934.20	3199.00 ± 2487.00	0.024 *
PlGF [pg/mL] (mean ± SD)	44.50 ± 32.23	71.70 ± 38.32	28.17 ± 9.56	<0.0001 **
sFlt-1/PlGF [median (range)]	44.24 (7.39–227.00)	22.72 (7.39–36.77)	76.00 (39.38–227.00)	<0.0001 **
sEng [ng/mL] (mean ± SD)	59.62 ± 51.40	44.24 ± 36.51	68.86 ± 57.26	0.131 ns
8-epi-PGF2α [pg/mL] [median (range)]	89.09 (10.98–531.20)	74.67 (10.98–304.30)	120.90 (19.06–531.20)	0.039 *
PlGF/sFlt-1 [median (range)]	0.023 (0.004–0.140)	0.040 (0.030–0.140)	0.013 (0.004–0.025)	<0.0001 **
sEng/PlGF [median (range)]	1.20 (0.16–8.41)	0.41 (0.16–1.86)	1.59 (0.46–8.41)	0.0003 *
8-epi-PGF2α/PlGF [median (range)]	2.27 (0.26–15.70)	1.45 (0.26–6.87)	3.87 (0.51–15.70)	0.003 *
Diagnosis at the time of inclusion				
Suspected PE [*n* (%)]	25 (62.50%)	10 (66.67%)	15 (60%)	-
Gestational hypertension [*n* (%)]	10 (25%)	-	10 (40%)	-
PE/HELLP syndrome [*n* (%)]	10 (25%)	3 (20%)	7 (28%)	-

ALT: alanine aminotransferase; AST: aspartate aminotransferase; DBP: diastolic blood pressure; HELLP syndrome: hemolysis, elevated liver enzymes, and low platelet count; PE: pre-eclampsia; SBP: systolic blood pressure; SD: standard deviation; sFlt-1: soluble fms-like tyrosine kinase-1; sEng: soluble endoglin; PlGF: placental growth factor; 8-epi-PGF2α: 8-epi-prostaglandin F2-alpha. Data from the pregnant women with suspected or confirmed PE were analyzed for statistical significance: * *p* < 0.05; ** *p* < 0.0001; ns = not significantly different.

**Table 2 biomedicines-12-00433-t002:** The dynamics of the mean respective to the median values of angiogenic and oxidative stress mediator levels during pregnancy follow-up in the two studied groups.

Parameter	sFlt-1/PlGF Ratio of ≤38 Group				sFlt-1/PlGF Ratio of >38 Group			
	18w	25w	32w	39w	18w	25w	32w	39w
sFlt-1 (mean)	848.00	1587.00	2519.00	3434.00	1352.00	2058.00	4684.00	3771.00
PlGF (mean)	50.40	73.61	75.88	54.26	24.90	17.48	19.18	16.68
sFlt-1/PlGF (median)	21.22	22.72	30.23	53.34	48.75	115.70	268.80	271.80
sEng (mean)	13.96	33.59	78.24	105.60	23.05	39.56	108.30	117.50
8-epi-PGF2α (median)	48.63	74.22	157.30	250.60	34.36	82.08	188.20	334.10
PlGF/sFlt-1 (median)	0.05	0.04	0.03	0.02	0.021	0.009	0.004	0.004
sEng/PlGF (median)	0.34	0.41	1.18	1.76	0.99	2.29	7.62	6.88
8-epi-PGF2α/PlGF (median)	0.87	0.93	3.10	6.97	1.76	7.12	9.87	17.37

PlGF: placental growth factor; sFlt-1: soluble fms-like tyrosine kinase-1; sEng: soluble endoglin; 8-epi-PGF2α: 8-epi-prostaglandin F2-alpha.

**Table 3 biomedicines-12-00433-t003:** Synopsis of correlations (Spearman’s test) between 8-epi-PGF2α and angiogenic mediator levels at baseline in the sFlt-1/PlGF ratio ≤38 group.

Parameter	sFlt-1	PlGF	sFlt-1/PlGF	sEng	8-epi-PGF2α
sFlt-1		rho = −0.754 *p* = 0.001 *	rho = 0.248 *p* = 0.373	rho = 0.466 *p* = 0.056 **	rho = 0.388 *p* = 0.053 **
PlGF			rho = −0.372 *p* = 0.172	rho = 0.194 *p* = 0.488	rho = 0.407 *p* = 0.132
sFlt-1/PlGF				rho = 0.374 *p* = 0.170	rho = 0.219 *p* = 0.046 *
sEng					rho = 0.330 *p* = 0.230

8-epi-PGF2α: 8-epi-prostaglandin F2-alpha; PlGF: placental growth factor; sFlt-1: soluble fms-like tyrosine kinase-1; sEng: soluble endoglin; * statistical significance; ** positive correlation to the limit of significance.

**Table 4 biomedicines-12-00433-t004:** Synopsis of correlations (Spearman’s test) between 8-epi-PGF2α and angiogenic mediator levels at baseline in the sFlt-1/PlGF ratio >38 group.

Parameter	sFlt-1	PlGF	sFlt-1/PlGF	sEng	8-epi-PGF2α
sFlt-1		rho = −0.650 *p* < 0.0001 *	rho = 0.856 *p* < 0.0001 *	rho = 0.669 *p* < 0.0001 *	rho = 0.724 *p* < 0.0001 *
PlGF			rho = −0.248 *p* = 0.232	rho = 0.285 *p* = 0.166	rho = 0.471 *p* = 0.147
sFlt-1/PlGF				rho = 0.306 *p* = 0.113	rho = 0.529 p = 0.037 *
sEng					rho = 0.675 *p* < 0.0001 *

8-epi-PGF2α: 8-epi-prostaglandin F2-alpha; PlGF: placental growth factor; sFlt-1: soluble fms-like tyrosine kinase-1; sEng: soluble endoglin; *: statistical significance.

## Data Availability

The data used to support the findings of this study are available from the corresponding author upon reasonable request.

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
