# Peer review of "Longitudinal 8-Epi-Prostaglandin F2-Alpha and Angiogenic Profile Mediator Evaluation during Pregnancy in Women with Suspected or Confirmed Pre-eclampsia"

_biomedicines, 2024, doi:10.3390/biomedicines12020433_

Round 1

Reviewer 1 Report

Comments and Suggestions for Authors

- A vizsgálatban a preeclampsiát vizsgáló kutatásból a már jól ismert sFlt-1-et, PlGF-et és sEng-t mérték. Újdonság, hogy ezek mellett meghatároztuk a 8-epi-PGF2α oxidatív stressz markert is, és a kapott adatokból különbözÅ‘ arányokat hasonlítottak össze. Ennek értékét csökkenti, hogy számos tanulmány vizsgálta az oxidatív stressz és a preeclampsia kapcsolatát.

- A vizsgálat komoly hiányossága, hogy nincs egészséges kontrollcsoport a vizsgálatban. Ezt ki kellene cserélni. Ily módon az eredmények potenciális prediktív értéke is bemutatható.

1. Az 1. ábrán hiányzik a Gyanús és MegerÅ‘sített PE mintaszáma, kérjük, töltse ki! Az ábra nem egységes (X számú beteg VAGY n=X), javítsák ki!

2. A kézirat csak a kezdeti klinikai adatokat tartalmazza (1. táblázat). Ezeket az értékeket a preeclampsia/gesztációs hypertonia diagnózisa után is szükséges lenne kimutatni. Pl. terhesség alatti maximális vérnyomásértékek, proteinuria, szülés hete, újszülött súlya...

3. Az 1. ábrán látható eredményekhez milyen szempontok szerint választották ki az egyéneket? Célszerű lenne az egyes ábrákat mediánértékekkel kiegészíteni.

4. Könnyebb lenne összehasonlítani más eredményekkel, ha más oxidatív stressz marker szintek is szerepelnének a kéziratban. Ha vannak még használható minták, javaslom a szabad tiol és/vagy malondialdehid meghatározását. Szerintem ezek mérése költséghatékony.

5. Az angiogén faktorok koncentrációja és az oxidatív stressz mértéke a preeclampsia súlyosságától is függhet. Szintén hasznos lenne a mért értékek összehasonlítása a közepesen súlyos és súlyos preeclampsiás csoportok között.

6. A 40 résztvevÅ‘t három csoportba osztották (PE, terhességi hipertónia, PE/HELLP szindróma). Ezeket a csoportokat nem vettük figyelembe az eredmények bemutatásakor, pedig eltérhettek egymástól. Javaslom a számított arányszámok táblázatos bemutatását ennek megfelelÅ‘en.

Author Response

Dear Reviewer,

Thank you very much for taking the time to analyze our manuscript, as well as for your kind appreciation and valuable suggestions.

All the typing recommended changes were performed in the body of our manuscript, with the Track Changes function activated.

Comments and Suggestions for Authors

- A vizsgálatban a preeclampsiát vizsgáló kutatásból a már jól ismert sFlt-1-et, PlGF-et és sEng-t mérték. Újdonság, hogy ezek mellett meghatároztuk a 8-epi-PGF2α oxidatív stressz markert is, és a kapott adatokból különbözÅ‘ arányokat hasonlítottak össze. Ennek értékét csökkenti, hogy számos tanulmány vizsgálta az oxidatív stressz és a preeclampsia kapcsolatát.

(In the study, the well-known sFlt-1, PlGF, and sEng were measured from the research investigating preeclampsia. What is new is that, in addition to these, the oxidative stress marker 8-epi-PGF2α was also determined, and different ratios were compared from the obtained data. The value of this is reduced by the fact that many studies have investigated the relationship between oxidative stress and preeclampsia.)

- A vizsgálat komoly hiányossága, hogy nincs egészséges kontrollcsoport a vizsgálatban. Ezt ki kellene cserélni. Ily módon az eredmények potenciális prediktív értéke is bemutatható.

(A serious shortcoming of the study is that there is no healthy control group in the study. This should be replaced. In this way, the potential predictive value of the results can also be presented.)

  1. Az 1. ábrán hiányzik a Gyanús és MegerÅ‘sített PE mintaszáma, kérjük, töltse ki! Az ábra nem egységes (X számú beteg VAGY n=X), javítsák ki!

The sample number of Suspected and Confirmed PE is missing in Figure 1, please fill it in! The diagram is not consistent (number of patients X OR n=X), correct it!)

  • Corrected

  1. A kézirat csak a kezdeti klinikai adatokat tartalmazza (1. táblázat). Ezeket az értékeket a preeclampsia/gesztációs hypertonia diagnózisa után is szükséges lenne kimutatni. Pl. terhesség alatti maximális vérnyomásértékek, proteinuria, szülés hete, újszülött súlya...

The manuscript contains only the initial clinical data (Table 1). These values should also be determined after the diagnosis of preeclampsia/gestational hypertension. E.g. maximum blood pressure values during pregnancy, proteinuria, week of delivery, newborn weight...)

  • We set out in this study to evaluate the dynamics of angiogenic markers and 8-epi-PGF2α, and highlight this aspect;
  • in subsection 3.2.1. ( Regarding the gestational age at which delivery occurred, in this group, it was between 37 weeks and 5 days and 39 weeks and 3 days.) and 3.2.2. (In this group, the gestational age at which delivery occurred, was between 37 weeks and 2 days and 39 weeks and 5 days.), we specified the gestational age at which delivery occurred.

  1. Az 1. ábrán látható eredményekhez milyen szempontok szerint választották ki az egyéneket? Célszerű lenne az egyes ábrákat mediánértékekkel kiegészíteni.

According to what criteria were the individuals selected for the results shown in Figure 1? It would be advisable to supplement the individual figures with median values.)

  • In Figure 2 are represented for each patient included in the study, in the two study groups (sFlt-1/PlGF ratio of ≤ 38 group, n=15 ; sFlt-1/PlGF ratio of > 38 group, n=25), the individual values determined during follow-up pregnancy; so we did not make a selection of patients;
  • - there are patients in whom these biomarkers were determined 2 times during follow-up pregnancy and are indicated as red or black points, or patients in whom these biomarkers were determined 3 times.
  • - being individual values, we cannot complete the figures with average values.
  • - We specified in text:

Also, Figure 2 shows the sFlt-1, PlGF, sFlt-1/PlGF ratio, sEng, 8-epi-PGF2α, and other ratios (PlGF/sFlt-1, sEng/PlGF, and 8-epi-PGF2α/PlGF) of individual patients during follow-up pregnancy.

Figure 2. Placental growth factor (PlGF) (A), soluble fms-like tyrosine kinase-1 (sFlt-1) (B), sFlt-1/PlGF ratio (C), soluble endoglin (sEng) (D), and 8-epi-prostaglandin F2-alpha (8-epi-PGF2α) (E) serum levels were measured in forty women who had sFlt-1/PlGF ratio of ≤38 (left) or >38 (right) at the presentation and different points in their pregnancies. Data for five patients for whom sFlt-1/PlGF ratio increased from ≤38 at the time of presentation to >38 later in pregnancy are indicated as red points (left).

  1. Könnyebb lenne összehasonlítani más eredményekkel, ha más oxidatív stressz marker szintek is szerepelnének a kéziratban. Ha vannak még használható minták, javaslom a szabad tiol és/vagy malondialdehid meghatározását. Szerintem ezek mérése költséghatékony.

It would be easier to compare with other results if other oxidative stress marker levels were included in the manuscript. If there are still usable samples, I recommend determining the free thiol and/or malondialdehyde. I think measuring them is cost-effective.)

  • In our study, we did not take into account the determination of free thiol and/or malondialdehyde, because in this second study, as in the first published one, we started from the findings:

8-iso-PGF2α is one of the isoprostanes that are easily detectable, persistent in bodily fluids, unaffected by food, and controlled by endogenous antioxidants.

 (37.     Boldeanu, L.; Văduva, C.-C.; Caragea, D.C.; Novac, M.B.; Manasia, M.; SiloÈ™i, I.; Manolea, M.M.; Boldeanu, M.V.; Dij-mărescu, A.L. Association between Serum 8-Iso-Prostaglandin F2α as an Oxidative Stress Marker and Immunological Markers in a Cohort of Preeclampsia Patients. Life 2023, 13, 2242. https://doi.org/10.3390/life13122242)

  1. Az angiogén faktorok koncentrációja és az oxidatív stressz mértéke a preeclampsia súlyosságától is függhet. Szintén hasznos lenne a mért értékek összehasonlítása a közepesen súlyos és súlyos preeclampsiás csoportok között.

The concentration of angiogenic factors and the degree of oxidative stress may also depend on the severity of preeclampsia. It would also be useful to compare the measured values between the moderately severe and severe preeclampsia groups.)

  • In this study, we did not aim to compare the measured values between the moderately severe and severe preeclampsia groups.
  • Perhaps, it is an objective for future studies.

  1. A 40 résztvevÅ‘t három csoportba osztották (PE, terhességi hipertónia, PE/HELLP szindróma). Ezeket a csoportokat nem vettük figyelembe az eredmények bemutatásakor, pedig eltérhettek egymástól. Javaslom a számított arányszámok táblázatos bemutatását ennek megfelelÅ‘en.

The 40 participants were divided into three groups (PE, gestational hypertension, PE/HELLP syndrome). We did not consider these groups when presenting the results, even though they may have differed from each other. I recommend the tabular presentation of the calculated ratios accordingly.)

  • In this study, we did not aim to compare the measured values between the three groups (PE, gestational hypertension, PE/HELLP syndrome).
  • It is an objective for future studies

Reviewer 2 Report

Comments and Suggestions for Authors

-    RECOMMENDATION: MINOR REVISION

The author performed an interesting exploratory study on the dynamic of angiogenic and oxidative stress markers in women with suspected or confirmed pre-eclampsia sampled at least two times in the same pregnancy.
Here are some considerations:
-    I would underline the longitudinal nature of the study adding this concept in the title, like “Longitudinal {name of the mediators} evaluation during pregnancy in women with suspected or confirmed pre-eclampsia” which is more of impact compared to the use of the word “dynamics”.
-    Please change in the Introduction page 2 line 61 : “later in life” instead of “in the future” and please add the following reference: Inversetti A, Pivato CA, Cristodoro M, Latini AC, Condorelli G, Di Simone N, Stefanini G. Update on long-term cardiovascular risk after pre-eclampsia: a systematic review and meta-analysis. Eur Heart J Qual Care Clin Outcomes. 2023 Nov 16:qcad065. doi: 10.1093/ehjqcco/qcad065. Epub ahead of print. PMID: 37974053.

-    Please add the following references in the introduction line 69 when you describe the relation between trophoblast cell dysfunction and adverse pregnancy outcomes:
o    Licini C, Avellini C, Picchiassi E, Mensà E, Fantone S, Ramini D, Tersigni C, Tossetta G, Castellucci C, Tarquini F, Coata G, Giardina I, Ciavattini A, Scambia G, Di Renzo GC, Di Simone N, Gesuita R, Giannubilo SR, Olivieri F, Marzioni D. Pre-eclampsia predictive ability of maternal miR-125b: a clinical and experimental study. Transl Res. 2021 Feb;228:13-27. doi: 10.1016/j.trsl.2020.07.011. Epub 2020 Jul 26. PMID: 32726711.

Author Response

Dear Reviewer,

Thank you very much for taking the time to analyze our manuscript, as well as for your kind appreciation and valuable suggestions.

All the typing recommended changes were performed in the body of our manuscript, with the Track Changes function activated.

Comments and Suggestions for Authors

RECOMMENDATION: MINOR REVISION

The author performed an interesting exploratory study on the dynamic of angiogenic and oxidative stress markers in women with suspected or confirmed pre-eclampsia sampled at least two times in the same pregnancy.

Here are some considerations:

-    I would underline the longitudinal nature of the study adding this concept in the title, like “Longitudinal {name of the mediators} evaluation during pregnancy in women with suspected or confirmed pre-eclampsia” which is more of impact compared to the use of the word “dynamics”.

  • Revised

-    Please change in the Introduction page 2 line 61 : “later in life” instead of “in the future” and please add the following reference: Inversetti A, Pivato CA, Cristodoro M, Latini AC, Condorelli G, Di Simone N, Stefanini G. Update on long-term cardiovascular risk after pre-eclampsia: a systematic review and meta-analysis. Eur Heart J Qual Care Clin Outcomes. 2023 Nov 16:qcad065. doi: 10.1093/ehjqcco/qcad065. Epub ahead of print. PMID: 37974053.

  • Revised
  • Added the reference

-    Please add the following references in the introduction line 69 when you describe the relation between trophoblast cell dysfunction and adverse pregnancy outcomes:

o    Licini C, Avellini C, Picchiassi E, Mensà E, Fantone S, Ramini D, Tersigni C, Tossetta G, Castellucci C, Tarquini F, Coata G, Giardina I, Ciavattini A, Scambia G, Di Renzo GC, Di Simone N, Gesuita R, Giannubilo SR, Olivieri F, Marzioni D. Pre-eclampsia predictive ability of maternal miR-125b: a clinical and experimental study. Transl Res. 2021 Feb;228:13-27. doi: 10.1016/j.trsl.2020.07.011. Epub 2020 Jul 26. PMID: 32726711.

  • Added the reference

Reviewer 3 Report

Comments and Suggestions for Authors

The authors present a manuscript which aims to invest 8-epi-prostaglandin F2-alpha and angiogenic mediators in pregnant women with suspected or confirmed preeclampsia. Although the study has been conducted properly and the manuscript has been well written. However, several corrections should be made to achieve better comprehension. First, the introduction part should be condensed into one page. Second, the authors should discuss what theirfindings add to what is already known about this subject (i.e., improving the diagnostics accuracy/predictive power of sFlt-1/PlGF or else?) in a separate paragraph of the discussion part. Third, the authors should mention about the factors that limit the power of their study in a separate paragraph of the discussion part.

I recommend that the revised version of this manuscript can be accepted for publication in Biomedicines. 

Author Response

Dear Reviewer,

Thank you very much for taking the time to analyze our manuscript, as well as for your kind appreciation and valuable suggestions.

All the typing recommended changes were performed in the body of our manuscript, with the Track Changes function activated.

Comments and Suggestions for Authors

The authors present a manuscript which aims to invest 8-epi-prostaglandin F2-alpha and angiogenic mediators in pregnant women with suspected or confirmed preeclampsia. Although the study has been conducted properly and the manuscript has been well written. However, several corrections should be made to achieve better comprehension.

First, the introduction part should be condensed into one page.

- Revised

Second, the authors should discuss what theirfindings add to what is already known about this subject (i.e., improving the diagnostics accuracy/predictive power of sFlt-1/PlGF or else?) in a separate paragraph of the discussion part.

-We formulated

  • In our opinion, this is the first study to evaluate the dynamics of the 8-epi-PGF2α at least two times during pregnancy. In our study, we determined in twenty-three of the forty cases even three measurements during the follow-up As in the case of sFlt-1, for the 8-epi-PGF2α we obtained in both groups, that the medians values have increased further in most cases, almost doubling roughly every measurement, with a mean change per day of 1.37 (sFlt-1/PlGF ratio ≤38 group), and 2.04 (sFlt-1/PlGF ratio >38), respectively.
  • The increase in oxidative stress interacts with angiogenic growth mediators and influences changes in placental angiogenesis in PE, which will unbalance the ratio be-tween angiogenic growth mediators and oxidative stress biomarkers among PE wom-en, an imbalance also found in our study. The combined evaluation of angiogenic growth mediators and oxidative stress biomarkers can be proposed and can contribute not only to diagnosis, prognosis, and therapeutic scope but also to a better understand-ing of the complexity of PE mechanisms. The ones exposed above are supported in our study by the strong and moderately significant correlations obtained between the 8-epi-PGF2α levels and the sFlt-1, sEng, and sFlt-1/PlGF ratio.
  • Following this exploratory study, we were able to observe that the dynamic de-termination of these biomarkers, at least two determinations during pregnancy, pro-vide us with important information about changes in the levels of the 8-epi-PGF2α and sEng, along with the well-known biomarkers, sFlt-1 and PlGF, during pregnancy.

Third, the authors should mention about the factors that limit the power of their study in a separate paragraph of the discussion part.

- We formulated

  • We recognize that the descriptive character of our study, the limited sample size, and the fact that the patients were recruited from a single clinical university hospital in Craiova provide certain limitations to the analysis.
  • The observations obtained in our study would certainly benefit from an additional extension of the study, with multicenter involvement. Further study on this area may yield valuable insights into identifying which serum biomarkers might serve as additional therapeutic targets in clinical practice.

Round 2

Reviewer 1 Report

Comments and Suggestions for Authors

The manuscript is acceptable in its current form. Where it has not been corrected, the reasons are acceptable.